# Antimicrobial Resistance Patterns and Risk Factors Associated with ESBL-Producing and MDR *Escherichia coli* in Hospital and Environmental Settings in Lusaka, Zambia: Implications for One Health, Antimicrobial Stewardship and Surveillance Systems

**DOI:** 10.3390/microorganisms11081951

**Published:** 2023-07-31

**Authors:** Maisa Kasanga, Geoffrey Kwenda, Jian Wu, Maika Kasanga, Mark J. Mwikisa, Raphael Chanda, Zachariah Mupila, Baron Yankonde, Mutemwa Sikazwe, Enock Mwila, Doreen M. Shempela, Benjamin B. Solochi, Christabel Phiri, Steward Mudenda, Duncan Chanda

**Affiliations:** 1Department of Epidemiology and Biostatistics, School of Public Health, Zhengzhou University, Zhengzhou 450001, Chinawujian@zzu.edu.cn (J.W.); 2Department of Biomedical Sciences, School of Health Sciences, University of Zambia, Lusaka 10101, Zambia; kwenda.geoffrey@unza.zm; 3Department of Pharmacy, University Teaching Hospital, Lusaka 50110, Zambia; maikakasanga@yahoo.com; 4Department of Pathology and Microbiology, University Teaching Hospital, Lusaka 50110, Zambiabenji.solochi@gmail.com (B.B.S.); 5Adult Centre of Excellence, University Teaching Hospital, Lusaka 50110, Zambia; 6Department of Pathology, Lusaka Trust Hospital, Lusaka 35852, Zambia; 7Churches Health Association of Zambia, Lusaka 34511, Zambia; 8Department of Laboratory and Research, Central University of Nicaragua, Managua 12104, Nicaragua; 9Department of Microbiology, School of Public Health, University of Zambia, Lusaka 10101, Zambia; 10Department of Pharmacy, School of Health Sciences, University of Zambia, Lusaka 10101, Zambia; 11Research and Surveillance Technical Working Group, Zambia National Public Health Institute, Lusaka 10101, Zambia

**Keywords:** *Escherichia coli*, ESBL, One Health, antibiotics, multidrug resistance, antimicrobial stewardship, Zambia

## Abstract

Antimicrobial resistance (AMR) is a public health problem threatening human, animal, and environmental safety. This study assessed the AMR profiles and risk factors associated with Escherichia coli in hospital and environmental settings in Lusaka, Zambia. This cross-sectional study was conducted from April 2022 to August 2022 using 980 samples collected from clinical and environmental settings. Antimicrobial susceptibility testing was conducted using BD Phoenix^TM^ 100. The data were analysed using SPSS version 26.0. Of the 980 samples, 51% were from environmental sources. Overall, 64.5% of the samples tested positive for *E. coli*, of which 52.5% were from clinical sources. Additionally, 31.8% were ESBL, of which 70.1% were clinical isolates. Of the 632 isolates, 48.3% were MDR. Most clinical isolates were resistant to ampicillin (83.4%), sulfamethoxazole/trimethoprim (73.8%), and ciprofloxacin (65.7%) while all environmental isolates were resistant to sulfamethoxazole/trimethoprim (100%) and some were resistant to levofloxacin (30.6%). The drivers of MDR in the tested isolates included pus (AOR = 4.6, CI: 1.9–11.3), male sex (AOR = 2.1, CI: 1.2–3.9), and water (AOR = 2.6, CI: 1.2–5.8). This study found that *E. coli* isolates were resistant to common antibiotics used in humans. The presence of MDR isolates is a public health concern and calls for vigorous infection prevention measures and surveillance to reduce AMR and its burdens.

## 1. Introduction

Antimicrobial resistance (AMR) is a growing public health crisis that affects both human and animal health and has a strong relationship with the environment [1,2,3,4,5]. The concerns regarding AMR have been steadily increasingworldwide, endangering a wide variety of effective medical interventions (e.g., surgery, chemotherapy, intensive care), and the ability to effectively prevent and cure infectious diseases [6,7]. It has been established by numerous studies that the overuse and misuse of antibiotics in the agricultural, veterinary, and human medical sectors promote the development and spread of multidrug-resistant (MDR) pathogens and allow for the emergence of novel resistance mechanisms [8,9,10,11]. Additionally, antimicrobial-resistant bacteria and their resistomes spread between humans, animals, and their environment [12,13,14]. In real-world settings, infections caused by MDR bacteria are associated with increased morbidity and mortality [4,6,7,15]. The significance of MDR infections has been estimated by the Global Burden of Disease (GBD) study, where it was shown that ~1.27 million (95% UI: 0.91–1.71 million) deaths were directly attributable to bacterial MDR globally in the year 2019 alone [6,16]. Among these MDR bacteria, *Escherichia coli* (*E. coli*), *Staphylococcus aureus*, *Klebsiella pneumoniae*, *Streptococcus pneumoniae*, *Acinetobacter baumannii,* and *Pseudomonas aeruginosa* were the most significant and associated with ~0.93 million (95% UI: 0.66–1.27 million) directly attributable deaths [6,7,16]. Alongside these consequences, other AMR implications include increased medical costs with a negative impact on the economy (both due to direct and indirect costs), and limited options to treat infections, endangering sustainable development globally [15,17,18,19,20,21].

*E. coli*, a gram-negative rod belonging to the Enterobacterales order, is a lactose and non-lactose fermenting microbe [22,23,24]. It is among the microorganisms that have developed considerable levels of resistance to most antimicrobials used in humans, animals, and agriculture, and has the potential to spread effectively in the environment [1,25,26,27,28,29,30,31]. The injudicious handling of antimicrobials in the One Health ecology further exacerbates the resistance situation [32,33,34,35,36,37]. *E. coli* is considered a major cause of pediatric infections that result in adverse outcomes [37,38]. It has been reported as one of the most common pathogens responsible for infections, particularly in countries with unstable healthcare and surveillance systems [6,39,40,41,42]. Despite being a common member of the intestinal microbiota in humans and animals, *E. coli* is also found in water, soil, and around plants, and is the leading cause of several common bacterial infections, including gastroenteritis, urinary tract infections (UTIs), septicemia, and neonatal meningitis [2,43,44,45,46,47,48,49]. In recent years, the rise of MDR *E. coli* has been documented in almost all countries worldwide [43,46,50,51,52,53]. The spread of extended-spectrum β-lactamase (ESBL)-producing bacteria (including ESBL-*E. coli*; ESBL-EC) through the environment—a major cause of healthcare problems (i.e., in healthcare-associated infections) and community-acquired infections—is caused by the increasing global dependence and use of β-lactam antibiotics (i.e., penicillins, cephalosporins, carbapenems, and monobactams), which necessitates urgent action [54,55,56,57].

ESBL-EC possesses various β-lactamase enzymes that rapidly evolve through the ability to hydrolyze antimicrobials and cause increased resistance to β-lactam antibiotics [57,58]. Extended-spectrum cephalosporins—such as cefotaxime, ceftriaxone, and ceftazidime—and monobactams (such as aztreonam) are susceptible to hydrolysis by ESBLs [59,60]. Resistance to other antimicrobial classes, such as aminoglycosides, macrolides, tetracyclines, quinolones, and sulfonamides, may be acquired by plasmid-encoded resistance determinants—coexisting in bacteria-harbouring ESBLs—rapidly reaching the phenotype of MDR, which further limits therapeutic options and poses a therapeutic conundrum [2,16,60,61]. β-lactam antibiotics are among the most commonly administered drugs globally, as they have an advantageous side effect profile and in many patient populations (i.e., children, the elderly, and pregnant women), they are the only suitable antimicrobials [62,63,64,65]. However, the development of resistance to these agents in recent years has become a serious public health concern [66,67,68]; this is especially true in low-income countries, such as Zambia, where β-lactam antibiotics are overused and misused and are often readily available without a medical prescription [63,64,69]. In addition to β-lactamases, the modification of penicillin-binding proteins (PBPs), the decreased permeability of the bacterial outer membrane, and the co-existence of several resistance mechanisms contributed to this phenomenon [57]. One of the direct causes of the development of ESBL strains in resource-constrained healthcare settings includes the empirical and symptomatic (without a diagnosis) use of antibiotics [70]. Despite multiple cases of nosocomial outbreaks attributed to these pathogens, there is limited information regarding the frequency of ESBL-producing bacteria in most Zambian hospitals.

Due to the isolation of MDR bacteria—such as *E. coli*—from humans, animals, and the environment, it is being increasingly understood that a One Health approach is required to address this problem [45,71,72]. This is because there is a clear interaction between humans and animals in the environment that can facilitate the transmission of *E. coli* from humans to animals or the environment and vice-versa [73]. Moreover, the presence of antimicrobial-resistant *E. coli* and other pathogens in humans, animals, and the environment calls for a holistic, multi-disciplinary collaborative action guided by the World Health Organization Global Action Plan (GAP) on AMR [74]. The One Health approach aims to address AMR across all the abovementioned domains (i.e., humans, animals, and the environment), as there is a higher transmission potential at the human and animal interface in the environment [1,5,75,76]. Therefore, a One Health approach (i.e., the systems thinking within ecological systems) promotes, and is an integral part of, antimicrobial stewardship (AMS) programmes for the prudent of antimicrobials in humans, animals, and the environment [77,78,79,80,81]. Most AMR data comes from high-income countries (HICs), while the AMR burden of sub-Saharan Africa (SSA)—including Zambia—is inadequately documented. In Zambia, the National Action Plan (NAP) on AMR was developed in 2017 in line with the GAP on AMR to tackle this problem using a One Health approach [82,83]. Alongside this, there have been some studies published to promote AMS in human and animal health in this geographical region [27,28,63,64,84,85,86,87,88,89,90,91]. However, there is still very little information on the isolation, resistance patterns, and risk factors associated with ESBL-producing and/or MDR *E. coli* originating in humans, food-producing animals, other food products, and the environment. With this in mind, this study aimed to comprehensively assess the AMR patterns and risk factors associated with ESBL-producing and MDR *E. coli* in hospital and environmental settings in Lusaka, Zambia.

## 2. Materials and Methods

### 2.1. Study Design and Site Location

The present cross-sectional study was conducted between April and August 2022 at the main referral University Teaching Hospital (UTH) and townships (administrative sub-districts) in the capital city of Zambia, Lusaka. The samples collected and included in the data analysis were (i) clinical samples (i.e., urine, stool, blood, cerebrospinal fluid) from inpatients and outpatients and (ii) environmental samples (i.e., meat, fruits and vegetables, water, and those isolated from hospital equipment). The UTH has a bed capacity of 1665 beds, acting as a national and the largest tertiary referral hospital in Lusaka that provides specialised patient care for patients from all over Zambia. Lusaka (360 km^2^) is the capital city of Zambia, with an estimated 687,923 households [92], and a human population of approximately 3,079,964 [93].

### 2.2. Data Collection

Data collection for clinical samples included the date and time of sample collection, sample type, anonymised identification code, and the age and sex of the patients. Information corresponding to the environmental samples included the source, type and area sampled. Only the samples kept at an ambient temperature for no longer than two hours were included in the study.

### 2.3. Specimen Collection and Processing

The environmental samples were first swabbed and enriched in buffered peptone water (BPW) (Oxoid Ltd., Basingstoke, Hampshire, UK) and incubated for 3 h at 37 °C. A sterile loop was dipped into the enriched BPW, where the sample was incubated. Afterwards, a 0.5 mL sample of incubated BPW was inoculated on CHROMagar™ ECC (*E. coli* and coliforms; HiMedia Laboratories Pvt. Ltd., Mumbai MS, India) agar plates at 37 °C for 18 to 24 h for the isolation of *E. coli*. Presumptive *E. coli* colonies were streaked on Eosin Methylene Blue (EMB) agar (Oxoid™, Basingstoke, Hampshire, UK) for the identification of *E. coli*.

For clinical specimens, the presumptive identification of *E. coli* colonies was defined as the growth of lactose-fermenting, donut-shaped colonies on Xylose Lysine Deoxycholate (XLD) agar (Oxoid Ltd., Basingstoke, Hampshire, UK), MacConkey agar (Oxoid Ltd., Basingstoke, Hampshire, UK), and Hichrome chromogenic UTI agar (HiMedia Laboratories Pvt. Ltd., Mumbai MS, India). Therefore, urine samples were inoculated directly onto Hichrome chromogenic UTI agar and incubated at 37 °C for 18 to 24 h. Presumptive *E. coli* colonies were characterised by the appearance of dark blue to violet colonies. The stool was inoculated and incubated on XLD for 24 h at 37 °C, and *E. coli* colonies were defined after the appearance of yellow colonies. Clinical specimens were inoculated directly on MacConkey agar (Oxoid Ltd., Basingstoke, Hampshire, UK) and incubated for 24 h at 37 °C. On MacConkey agar, lactose-fermenting colonies appeared pink in colour while non-lactose-fermenting colonies appeared off-white opaque. On EMB, greenish metallic colonies were presumed to be *E. coli* and were sub-cultured on nutrient agar (Oxoid Ltd., Basingstoke, Hampshire, UK), where they appeared large, thin, circular, and greyish-white after 24 h of aerobic incubation at 37 °C. To differentiate *E. coli* from other lactose-fermenting bacteria, phenotypic confirmation was performed on all pure colonies using a battery of biochemical tests, including triple sugar iron (TSI) agar, lysine iron agar (LIA), simmons citrate agar (SCA), and sulfide indole motility (SIM) agar, respectively. Only colonies that passed the biochemical tests were identified as *E. coli*. The identified presumptive colonies of *E. coli* were selected and cultured on nutrient agar for purification purposes and further analysis. For further confirmation, *E. coli* isolates were subjected to identification using the Becton Dickinson BD Phoenix^TM^ 100 system (BD Diagnostic Systems, Sparks, MD, USA).

### 2.4. Antibiotic Susceptibility Testing

Antimicrobial susceptibility testing (AST) for the respective *E. coli* isolates was performed using disk diffusion and the BD Phoenix^TM^ 100 Automated Microbiology System (BD Diagnostic Systems, Sparks, MD; based on minimum inhibitory concentrations). The following antibiotics were used for AST: ampicillin 10 µg (AMP), amoxicillin/clavulanic acid 10 µg (AMC), cefepime 30 µg (FEP), ceftazidime 30 µg (CAZ), cephazolin 30 µg (KZ), ceftriaxone 30 µg (CRO), cefuroxime 30 µg (CXM), ciprofloxacin 5 µg (CIP), ertapenem 30 µg (ETP), gentamicin 10 µg (CN), imipenem 10 µg (IPM), levofloxacin 5 µg (LEV), nitrofurantoin 30 µg (NIT), and sulfamethoxazole/trimethoprim 23 µg (SXT). Interpretation of the AST results (i.e., defined as susceptible, intermediate or resistant) was based on the standards and breakpoints as defined by the Clinical and Laboratory Standard Institute (CLSI) [94]. Furthermore, ESBL-producing isolates were confirmed by the combined double-disk test (with cefotaxime and ceftazidime alone, and in combination with cefotaxime/clavulanic acid) and the Becton Dickinson BD Phoenix^TM^ 100 system (Becton, Dickinson Company, Sparks, MD, USA) as defined by CLSI guidelines [94]. Each batch incorporated a control strain of *E. coli* ATCC 25922 to ensure the validity and reliability of AST. Isolates were classfied as MDR (resistance to at least one agent in ≥3 different antibiotic classes), extensive drug resistance (XDR; susceptibility to 1 or 2 remaining antibiotics), and pan-drug resistance (non-susceptiblity to all classes of antibiotics) (PDR) [95].

### 2.5. Data Analysis

The raw data of the isolates was summarised, cleaned, and coded in Microsoft Excel 2013 (Microsoft Corp., Redmond, WA, USA). Descriptive analysis was conducted to characterise the data using means, medians, ranges, and percentages. Various statistical tests were employed to determine the factors associated with ESBL and MDR *E. coli* isolates, including Chi-square tests, univariate and multiple logistic regression (ESBL), and multinomial (MDR) analyses. The backward elimination method (MDR: based on the likelihood ratio test) was utilised to select the most relevant variables, accounting for confounding factors. Adherence to the assumptions of the Chi-square tests was ensured, and if not met, Fisher’s exact test with Monte Carlo simulation (*n* = 1000) was used. The analyses were performed using the Statistical Package for Social Sciences (SPSS), version 26.0 (IBM Corp, Armonk, NY, USA). The normality of the data was assessed through the Kolmogorov-Smirnov test. All statistical tests were performed at a 95% confidence level with a *p* < 0.05 indicating statistical significance.

## 3. Results

### 3.1. Descriptive Characteristics of Clinical and Environmental E. coli Strains

A total of *n* = 980 samples were collected and subjected to microbial culture for *E. coli* using phenotypic methods. Of the *n* = 980 samples, *n* = 480 were from clinical sources, while *n* = 500 were from environmental sources (Figure 1); out of the total sample number, *E. coli* was isolated from *n* = 632 (64.5%) of samples, where the distribution was *n* = 332 (69.2%) and *n* = 300 (60.0%) from clinical and environmental sources, respectively.

The characteristics of patients and specimens corresponding to positive clinical samples (*n* = 332) are summarised in Table 1. Most of the samples were from female participants (58.7%), patients aged 0 to 14 years, urine (74.4%), and outpatient department (35.5%) (Table 1).

The origins of the *n* = 300 environmental *E. coli* specimens are summarised in Table 2. The majority of the samples were from medical equipment, meat and fruits/vegetables (Table 2).

### 3.2. Antibiotic Susceptibility Patterns of E. coli Isolated from Clinical Samples

The majority of the clinical *E. coli* isolates were highly resistant to AMP, SXT, CIP, KZ, and LEV (Table 3). However, the isolates were highly susceptible to ETP, IPM, NIT, and CN. Higher rates of resistance in clinical *E. coli* strains were shown against penicillin-derivatives, fluoroquinolones, cephalosporins, and SXT in specimens such as blood cultures, CSF, and urine; additionally, only two that were resistant to carbapenems were from pus samples (Appendix A). Most of the *E. coli* isolates with extensive resistance originated from general adults’ wards, the ICU (both adult and neonatal), surgical wards, and the pediatric unit; the isolates that were resistant to CL were from the adult medical ward (*n* = 1), the outpatient department (*n* = 1), and from paediatrics (*n* = 2); while the isolates that were resistant to carbapenems were from the outpatient department (*n* = 1) and surgical unit (*n* = 1), respectively (Appendix A).

### 3.3. Antibiotic Susceptibility Patterns of E. coli Isolated from Environmental Samples

Isolates of *E. coli* from environmental samples were highly resistant to SXT, followed by LEV and KZ. However, the isolates were highly susceptible to ETP, IPM, CN, AMP, and CRO (Table 4). Environmental *E. coli* showing higher rates of non-susceptibility was isolated from water, fruits/vegetables and medical equipment (Appendix A).

### 3.4. Prevalence of ESBL-Producing, and MDR/XDR E. coli from Clinical and Environmental Sources

Overall, 48.3% (*n* = 304/632) of *E. coli* were MDR (clinical: 67.4% [*n* = 205/304], environmental 32.5% [*n* = 99/304]), while 13.2% (*n* = 40/304) were XDR (clinical: 32.5% [*n* = 13/40], environmental 67.5% [*n* = 27/40]); MDR isolates were more common among *E. coli* from clinical sources (*p* = 0.021), while this association was not found for XDR isolates (*p* = 0.729). The detailed distribution of MDR and XDR *E. coli* among environmental and clinical samples is presented in Appendix A. The overall prevalence of ESBL-EC was 31.8% (*n* = 201), out of which, 70.1% (*n* = 141) were of clinical, while 29.9% *n* = 60 were of environmental origin; ESBL-EC were significantly more common in clinical than environmental samples (*p* = 0.0328).

The largest number of ESBL-EC were from samples of patients aged between 0 and 14 years, females (54.6%), urine (56.7%), pus (34%), outpatient department (27.7%), and medical equipment (43.4%) (Table 5). Statistical significance was found among isolates from CSF, urine, surgical ward, and meat (Table 5).

This study found that isolates from samples of individuals aged between 45 and 54 years (AOR = 0.175, CI: 0.047–0.651) were less likely to be ESBL-EC compared to those aged between 0 and 14 years. Additionally, isolates from CSF were less likely to be ESBL-EC (AOR = 0.050, CI: 0.005–0.363) compared to those from blood. Finally, isolates from urine were less likely to be ESBL-EC (AOR = 0.093, CI: 0.014–0.388) compared to those from blood (Table 6).

Most MDR *E. coli* were isolated from samples of patients aged between 0 and 14 years (24.9%), males (52.7%), urine (66.3%), outpatient department (29.3%), and fruits/vegetables (44.4%). This study revealed that MDR *E. coli* isolates were significantly associated with age, sex, specimen type, hospital department, and environmental samples (Table 7).

Table 8 summarises the results of the logistic regression analysis for the factors significantly associated with MDR in *E. coli* isolates: notably, clinical isolates originating from pus and male patients were significantly associated with the MDR phenotype; in the case of environmental sources, isolates from water were significantly associated with the MDR phenotype (Table 8).

## 4. Discussion

AMR has become a rising global burden, endangering global public health and sustainable healthcare for both developing and developed countries [6,7]. According to the O’Neill report, AMR may become the second leading cause of death by 2050, responsible for over 10 million deaths worldwide [96]. To establish effective regional, national, and global strategies to curb AMR, it is essential to investigate the prevalence of this problem and to develop empirical treatment strategies (local antibiograms). The objective of the present study was to examine ESBL-producing *E. coli* and the antimicrobial susceptibility patterns of isolates from various clinical and environmental sources to a wide range of antibiotic groups. Production of β-lactamase enzymes—especially ESBLs, owing to their rapid and successful spread across the globe, is one of the most significant mediators conferring resistance to a wide range of β-lactams in *E. coli* [43,97]. These enzymes form a large class of resistance determinants that are frequently encoded on plasmids and are a major driver in the emergence of MDR that confers resistance to penicillins and cephalosporins. Due to their considerable prevalence, clinicians are now often forced to use carbapenems (the last of the β-lactams), or other antibiotics with more disadvantageous adverse effects [98].

This present study investigated the AMR profiles and risk factors associated with ESBL-producing *E. coli* in hospital and environmental settings in Zambia. This study found that the prevalence of *E. coli* was 64.5%, of which 52.5% were from clinical sources. Additionally, 31.8% were ESBL, of which 70.1% were clinical isolates. Of the 632 isolates, 48.3% were MDR. Most clinical isolates were resistant to AMP (83.4%), SXT (73.8%), and CIP (65.7%) while most environmental isolates were resistant to SXT (100%). The risk factors associated with MDR of the tested *E. coli* isolates included pus, male sex, and water. Finally, *E. coli* isolates from samples of patients aged from 45 to 54 years and urine were less likely to be ESBL-producers.

The present study found the prevalence of *E. coli* to be 64.5%, of which 52.5% were isolated from clinical samples and 47.5% from environmental samples. Our low prevalence of *E. coli* isolated from the environment compared to the hospital setting could be due to the challenges in isolation methods of *E. coli* from environmental samples [99]. The prevalence of *E. coli* found in our study is higher than that reported in Pakistan where 23.75% of *E. coli* were isolated from urine samples [100]. A study in Poland reported a higher *E. coli* isolation rate of 78% (identified using 16S rRNA sequencing) and 82% (identified using MALDI Biotype) from river water and wastewater [100]. However, our isolation rate of *E. coli* from water samples was lower compared to that reported in Pakistan where the researchers found an isolation rate of 26.7% [100]. These differences in isolation rates could be due to technical differences and slight variability in methods. Interestingly, the isolation of *E. coli* from environmental and clinical samples demonstrates the need for a One Health approach in the surveillance of infections and AMR [73,101]. Additionally, genomic surveillance of priority pathogens should be promoted [102].

The current study found that most clinical *E. coli* isolates were highly resistant to AMP. Our findings corroborate findings from other studies where *E. coli* isolates from clinical and environmental samples were highly resistant to penicillins such as AMP [85,103,104,105,106,107,108,109]. The high resistance of *E. coli* to ampicillin can be attributed to its potential to develop intrinsic resistance against penicillins. Additionally, exposure to antibiotics such as penicillins also contributes to the high resistance of *E. coli* reported in many studies [47,48,50,85,89,91,110]. Conversely, a lower resistance rate of *E. coli* to ampicillin has been reported in other studies’ findings. The lower resistance can be due to the low use of antibiotics in other settings. Other studies found a high resistance of *E. coli* isolated from clinical and environmental samples to SXT [103,109,111,112,113]. The high resistance of *E. coli* to SXT could be due to its overuse and misuse in humans and animals [85,114,115,116]. However, a low resistance rate of *E. coli* to SXT was reported in a study that was conducted in Turkey among outpatients [117]. Similarly, low resistance of *E. coli* to SXT was reported in Australia due to the restriction of antibiotic use [118]. Hence, restricting the use of antibiotics may help curb AMR [119,120].

Our study also found high resistance of *E. coli* to quinolones such as CIP and LEV. The high resistance of *E. coli* to quinolones has been reported in other studies [114,121,122,123,124,125,126]. This high resistance could be due to the overuse and misuse of quinolones in human and animal health systems [114]. This is a huge problem because quinolones are largely used to treat urinary tract infections, respiratory tract infections, and other infections. However, lower resistance rates of *E. coli* to quinolones have been reported in similar settings [103,127]. This low use could be due to the effective implementation of AMS programmes in healthcare facilities. High resistance of *E. coli* to cephalosporins such as cefuroxime was found in our study. This is similar to a study that was conducted in Uganda and Nigeria, where *E. coli* isolates were 100% resistant to cefuroxime [109,125], and in South Africa where high resistance of *E. coli* to cephalothin was reported [128]. High resistance to Ceftriaxone [129,130], Ceftazidime [125], and Nalidixic acid [125] has also been reported. In Zambia, there is an overuse and misuse of antibiotics such as cephalosporins, which could be a driver of the high resistance [63,64,90,131].

The present study found that *E. coli* isolates were highly susceptible to CN, ETP, IPM, and NIT. This was observed for both clinical and environmental isolates. These findings corroborate reports from a meta-analysis where *E. coli* was highly susceptible to antibiotics such as amikacin, IPM, and NIT [103]. High susceptibility of *E. coli* to gentamicin was also reported in South Africa [128] and other similar studies [106,132]. The high susceptibility of *E. coli* isolates to IPM was also reported in other studies [129,133,134]. The high susceptibility of *E. coli* to these antibiotics suggests that they are the most effective drugs for the treatment of infections caused by *E. coli,* such as UTIs [132].

The current study found that 48.3% of *E. coli* isolates were MDR. A comparable *E. coli* MDR prevalence of 49.48% was reported in Ghana [135]. However, the finding in our study is lower than the 52% MDR reported in South Africa [128], 63.3% in Mexico [136], 68.3% in Ethiopia [43], 80% in Brazil [137], 91.4% in the United Kingdom [133], 97% in another Mexican study [138], and 98% in Bangladesh [139]. It is well known that susceptibility patterns can change over time and can differ between geographical locations [140]. Further, the high MDR among the *E. coli* isolates in hospital and environmental settings is partially due to the misuse and overuse of antibiotics both in humans and the environment [141]. It is also critical to note that MDR pathogens limit antibiotic treatment options, contributing to increased morbidity and mortality globally [141]. Therefore, the study of bacterial resistance to multiple antibiotics is essential for determining the most effective therapy for the subsequent infection, as the rise of MDR bacterial strains poses a significant threat to the health of people of all ages.

ESBL-producing *E. coli* may arise from interactions between ESBL type, strain genetic background, and selective pressures in various ecologic niches [54,142,143,144,145]. ESBL-producing *E. coli* is an important cause of both nosocomial and community-onset infections globally [146]. Additionally, ESBL-producing *E. coli* often shows resistance to multiple drugs, which limits treatment options [56,147,148,149]. Commonly used treatments for severely ill patients, such as fluoroquinolones, aminoglycosides, and trimethoprim, are often associated with co-resistance, resulting in higher rates of morbidity and mortality [150].

In this present study, the prevalence of ESBL-producing *E. coli* was found to be 31.8%. A low prevalence of ESBL-producing *E. coli* was reported in other studies [58,151,152]. Consequently, a higher prevalence of ESBL-producing *E. coli* was reported in other studies, including 38% in Sudan [124], 38.07% in China [153], 42.5% in Thailand [154], 50% in Brazil [137], 55.5% in India [134], 57.7% in Ethiopia [43], 62% in Jordan [155], and 88.8% in the United Kingdom [133]. The overuse and misuse of antibiotics, especially cephalosporins and fluoroquinolones in humans, animals, and the environment, have contributed to the emergence of ESBL-producing *E. coli* [153]. The increased ESBL-producing *E. coli* indicates a greater extent of resistance to antibiotics. Consequently, increased rates of ESBL producers limit treatment options [156].

The present study found that most ESBL producers were isolated from urine (56.7%). This finding is different from a study that was conducted in India that found that most of the ESBL-producing strains were isolated from blood (66.67%) [134]. Further, our study revealed that most ESBL producers were isolated from the outpatient department, in contrast with findings from a similar study where most ESBL producers were isolated from in-patients [134]. A study in the United Arabs Emirates (UAE) reported that ESBL-producing *E. coli* were responsible for 75% of UTIs in communities, indicating their high prevalence in outpatients [157]. Our study revealed that *E. coli* isolates from samples of patients aged between 45 and 54 years, CSF, and urine were less likely to be ESBL-producers. Older age was found to be a risk factor for ESBL-producing *E. coli* [158]. Similar studies have reported other risk factors of ESBL-producing *E. coli,* including previous hospitalisations, and use of urinary catheters [155].

In this study, most MDR *E. coli* isolates were isolated from samples of patients aged between 0 and 14 years, males, urine, outpatient department, and fruits/vegetables. The isolation of MDR *E. coli* from similar samples has been reported in other studies [159]. Additionally, *E. coli* isolates from males were more likely to be MDR than those from female patients. This is in line with other studies that reported similar results of males having higher odds of harbouring MDR *E. coli* isolates than females [160,161]. The impact of sex on the pattern of resistance was solely dependent on the clinical factors and location of the samples within the clinical isolates. Further, the risk of isolating MDR *E. coli* in our study was noted from pus samples. Our findings are similar to reports from previous studies which reported a larger fraction of MDR *E. coli* from pus [162,163]. However, some studies revealed that urine had a high prevalence of MDR *E. coli* [164,165,166,167,168]. Additionally, the present study found that water (drinking water from the community taps, boreholes, and wells) was significantly associated with MDR. This may be due to contaminated water sources within the communities or poor water quality. This is similar to a study in Zambia that reported that shallow water in peri-urban areas was significantly more contaminated with *E. coli* [169]. Our findings conform to other studies that have demonstrated the presence of high rates of MDR *E. coli* in water samples [170,171,172,173,174,175]. These findings indicate the presence of MDR *E. coli* in various samples.

We are aware of the limitations of our study: This study was conducted in the Lusaka province of Zambia; therefore, generalisation of the findings should be performed with caution. Additionally, we did not collect equal numbers of clinical and environmental samples, which may affect the comparison of results. However, we believe that the obtained results on the AMR patterns of *E. coli* isolated from clinical and environmental settings require heightened surveillance programs. Additionally, the identified risk factors, including isolates from pus, male sex, and water samples, emphasise the need for a One Health approach, which is critical to the surveillance of AMR across the human, animal, and environmental sectors.

## 5. Conclusions

This study reported a high prevalence rate of ESBL-producing *E. coli* among clinical and environmental samples. Most of these *E. coli* strains showed multiple AMR patterns to commonly used antibiotics, most of which were MDR and potential XDR strains. Significantly, risk factors in ESBL strains were associated with pus and blood specimens, with most isolates showing high resistance to cephalosporins, fluoroquinolones, ampicillin, and colistin, and only a few isolates being sensitive to aminoglycosides and carbapenems. The importance of these findings was the identification of ESBL-producing *E. coli* in humans, animals, and the environment. This suggests that surveillance and routine screening for MDR and ESBL-producing *E. coli* is important to control the spread of resistant strains as part of a One Health approach.

## Figures and Tables

**Figure 1 microorganisms-11-01951-f001:**
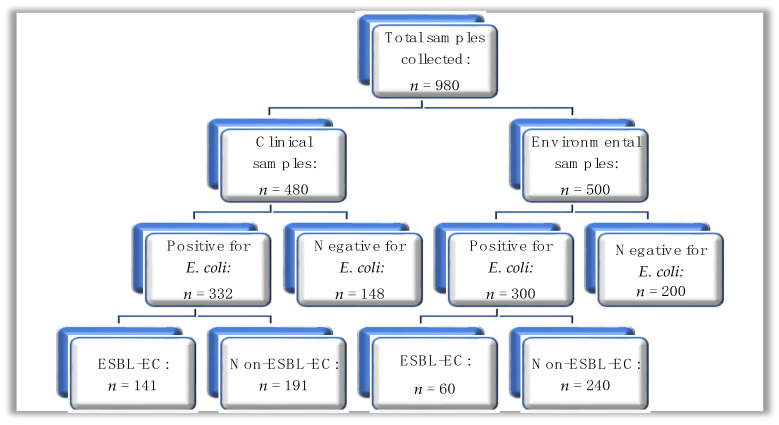
A hierarchical diagram showing the summary of clinical and environmental samples processed for *E. coli*; ESBL-EC: ESBL-producing *E. coli*.

**Table 1 microorganisms-11-01951-t001:** Descriptive characteristics of clinical samples positive for *E. coli*.

Variables	Frequency (*n*)	Percentages (%)
Sex		
Female	195	58.7
Male	137	41.3
Age (years)		
0–14	69	20.8
15–24	54	16.3
25–34	74	22.3
35–44	34	10.2
45–54	34	10.2
55 and above	67	20.2
Specimen Type		
Blood	12	3.6
Cerebrospinal fluid (CSF)	11	3.3
Pus	60	18.1
Stool	2	0.6
Urine	247	74.4
Origin of the sample (hospital department)		
Admission	8	2.4
General adult	52	15.7
Intensive care unit (ICU)	21	6.3
Obstetrics and Gynaecology	38	11.4
Outpatient Department (OPD)	118	35.5
Paediatrics and Neonatology	45	13.6
Surgery	50	15.1

**Table 2 microorganisms-11-01951-t002:** Descriptive characteristics of samples positive for *E. coli* from environmental sources.

Environmental Samples	Frequency (*n*)	Percentages (%)
Chicken and eggs	9	3.0
Fish	29	9.7
Water	35	11.7
Meat	56	18.7
Fruits and Vegetables	65	21.7
Medical Equipment	106	35.3

**Table 3 microorganisms-11-01951-t003:** Antibiotic susceptibility patterns of *E. coli* isolated from clinical samples.

Antibiotic Categories	Antibiotics	*n* (%)		
		Susceptible	Intermediate	Resistant
Aminoglycosides				
	CN	281 (84.6%)	7 (2.1%)	44 (13.3%)
Carbapenems	ETP	326 (98.2%)	4 (1.2%)	2 (0.6%)
	IPM	328 (98.8%)	2 (0.6%)	2 (0.6%)
Cephalosporins	KZ	113 (34%)	4 (1.2%)	215 (64.8%)
	CXM	117 (35.2%)	8 (2.4%)	207 (62.4%)
	CAZ	157 (47.3%)	9 (2.7%)	166 (50%)
	CRO	123 (37%)	-	209 (63%)
	FEP	159 (47.9%)	-	173 (52.1%)
Penicillin-derivatives	AMP	55 (16.6%)	-	277 (83.4%)
	AMC	118 (35.5%)	98 (29.6%)	116 (34.9%)
Sulphonamides	SXT	87 (26.2%)	-	245 (73.8%)
Furans	NIT	298 (89.8%)	18 (5.4%)	16 (4.8%)
Fluoroquinolones	CIP	113 (34%)	1 (0.3%)	218 (65.7%)
	LEV	119 (35.8%)	-	213 (64.2%)

Abbreviations: AMC-Amoxicillin/clavulanic acid; FEP-Cefepime; CN-Gentamicin; AMP-Ampicillin; IMP-Imipenem; CAZ-Ceftazidime; KZ-Cephazolin; CIP-Ciprofloxacin; LEV-Levofloxacin; NIT-Nitrofurantoin; CRO-Ceftriaxone; CXM-Cefuroxime; SXT-Sulfamethoxazole/trimethoprim; ETP-Ertapenem.

**Table 4 microorganisms-11-01951-t004:** Antibiotic susceptibility patterns of *E. coli* isolated from environmental samples.

Antibiotic Categories	Antibiotics	*n* (%)		
		Susceptible	Intermediate	Resistant
Penicillin	AMP	241 (80.3%)	-	59 (19.7%)
	AMC	238 (79.3%)	9 (3%)	53 (17.7%)
Cephalosporins	KZ	215 (71.7%)	-	85 (28.3%)
	CXM	235 (78.3%)	4 (1.3%)	61 (20.4%)
	CAZ	243 (81%)	-	57 (19%)
	CRO	241 (80.3%)	-	59 (19.7%)
	FEP	246 (82%)	-	54 (18%)
Carbapenems	ETP	300 (100%)	-	-
	IPM	300 (100%)	-	-
Aminoglycosides	CN	244 (81.3%)	2 (0.7%)	54 (18%)
Fluoroquinolones	CIP	225 (75%)	-	75 (25%)
	LEV	200 (66.7%)	8 (2.7%)	92 (30.6%)
Furans	NIT	237 (79%)	4 (1.3%)	59 (19.7%)
Sulphonamides	SXT	-	-	300 (100%)

**Table 5 microorganisms-11-01951-t005:** Epidemiological characteristics of ESBL-positive and negative *E. coli* among clinical and environmental isolates.

Variables	ESBL Positive		Total (*n*)			
	Negative *n* (%)	Positive *n* (%)		OR	95%CI	*p*-Value
Clinical Samples						
Age (Years)						
0–14	33 (17.3%)	36 (25.5%)	69	-	-	-
15–24	35 (18.3%)	19 (13.5%)	54	0.498	0.239–1.034	0.0615
25–34	41 (21.5%)	33 (23.4%)	74	0.738	0.382–1.425	0.3652
35–44	20 (10.5%)	14 (9.9%)	34	0.642	0.280–1.472	0.2950
45–54	22 (11.5%)	12 (8.5%)	34	0.500	0.214–1.167	0.1088
55 and above	40 (20.9%)	27 (19.1%)	67	0.619	0.314–1.220	0.1660
Sex						
Female	118 (61.8%)	77 (54.6%)	195	-	-	-
Male	73 (38.2%)	64 (45.4%)	137	0.744	0.479–1.158	0.1901
Specimen Type						
Blood	2 (1%)	10 (7.1%)	12	-	-	-
Cerebrospinal fluid (CSF)	8 (4.2%)	3 (2.1%)	11	0.075	0.010–0.563	0.0118
Pus	12 (6.3%)	48 (34%)	60	0.800	0.154–4.144	0.7904
Urine	169 (88.5%)	80 (56.7%)	249	0.095	0.021–0.448	0.0027
Origin of the sample (hospital department)						
Admission/Adult	40 (20.9%)	20 (14.2%)	60	-	-	-
Intensive care unit (ICU)	9 (4.7%)	12 (8.5%)	21	2.667	0.964–7.375	0.0588
Obstetrics and Gynecology	22 (11.5%)	16 (11.3%)	38	0.987	0.510–1.910	0.9698
Outpatient Department (OPD)	79 (41.4%)	39 (27.7%)	118	1.454	0.629–3.364	0.3811
Paediatrics and Neonatology	23 (12%)	22 (15.6%)	45	1.913	0.865–4.230	0.1091
Surgery	18 (9.4%)	32 (22.7%)	50	3.555	1.616–7.821	0.0061
Environmental Samples						
Fish	23 (8.1%)	6 (8.7%)	29	-	-	-
Water	22 (7.7%)	13 (18.8%)	35	2.265	0.732–7.014	0.156
Meat	62 (21.8%)	3 (4.3%)	65	0.185	0.043–0.804	0.024
Fruits and Vegetables	48 (16.8%)	17 (24.6%)	65	1.358	0.473–3.900	0.570
Medical Equipment	130 (45.6%)	30 (43.5%)	160	0.885	0.331–2.362	0.807

Note: SE = Standard error; OR = Odds ratio; B = Beta Estimate; CI = Confidence interval.

**Table 6 microorganisms-11-01951-t006:** Risk factors associated with ESBL-producing *E. coli*.

Variables		B (SE)	*p*-Value	AOR	95% CI for AOR	
					Lower	Upper
Sex	Male	−0.6270 (0.5805)	0.280	0.534	0.171	1.666
Age	15–24	−0.4757 (0.7081)	0.5017	0.621	0.155	2.489
	25–35	1.2354 (0.7572)	0.1028	3.440	0.780	15.173
	35–44	0.2458 (0.7653)	0.7481	1.279	0.285	5.730
	45–54	−1.7412 (0.6694)	0.0093	0.175	0.047	0.651
	>55	−0.9926 (0.5248)	0.0586	0.371	0.132	1.037
Sample	CSF	−2.9888 (1.0846)	0.0059	0.050	0.005	0.363
	Pus	−0.0740 (0.880)	0.9336	0.929	0.123	0.4631
	Urine	−2.3720 (0.8148)	0.0036	0.093	0.014	0.388

Note: AOR = Adjusted Odds Ratio.

**Table 7 microorganisms-11-01951-t007:** Distribution of MDR *E. coli* isolates among clinical and environmental samples.

Variables	Total (*n*)	MDR				
		Negative *n* (%)	MDR *n* (%)	XDR *n* (%)	Chi-Square	*p*-Value
Clinical Samples						
**Age (Years)**					22.900	0.004
0–14	69 (20.3%)	17 (14.9%)	51 (24.9%)	1 (7.7%)		
15–24	54 (16.3%)	25 (21.9%)	26 (12.7%)	3 (23.1%)		
25–34	74 (22.3%)	26 (22.8%)	47 (22.9%)	1 (7.7%)		
35–44	34 (10.2%)	17 (14.9%)	14 (6.8%)	3 (23.1%)		
45–54	34 (10.2%)	14 (12.3%)	18 (8.8%)	2 (15.4%)		
55 and above	67 (20.2%)	15 (13.2%)	49 (23.9%)	3 (23.1%)		
**Sex**					11.083	0.004
Female	137 (41.3%)	33 (28.9%)	97 (47.3%)	7 (53.8%)		
Male	195 (58.7%)	81 (71.1%)	108 (52.7%)	6 (46.2%)		
**Specimen Type**					41.905	<0.001
Blood	12 (3.6%)	0 (0.0%)	10 (4.9%)	2 (15.4%)		
Cerebrospinal fluid (CSF)	11 (3.3%)	0 (0.0%)	10 (4.9%)	1 (7.7%)		
Pus	60 (18.1%)	8 (7.0%)	49 (23.9%)	3 (23.1%)		
Stool	2 (0.6%)	2 (1.8%)	0 (0.0%)	0 (0.0%)		
Urine	247 (74.4%)	104 (91.2%)	136 (66.3%)	7 (53.8%)		
**Hospital Department**					25.464	0.005
Admission	8 (2.4%)	6 (5.3%)	2 (1.0%)	0 (0.0%)		
General adult	52 (15.7%)	13 (11.4%)	35 (17.1%)	4 (30.8%)		
Intensive care unit (ICU)	21 (6.3%)	3 (2.6%)	18 (8.8%)	0 (0.0%)		
Obstetrics and Gynecology	38 (11.4%)	15 (13.2%)	22 (10.7%)	1 (7.7%)		
Outpatient Department (OPD)	118 (35.5%)	54 (47.4%)	60 (29.3%)	4 (30.8%)		
Paediatrics and Neonatology	45 (13.6%)	12 (10.5%)	32 (15.6%)	1 (7.7%)		
Surgery	50 (15.1%)	11 (9.6%)	36 (17.6%)	3 (23.1%)		
**Environmental Samples**						
Fish	18 (8.8%)	7 (5.7%)	4 (14.8%)	29 (8.2%)	18.037	0.001
Water	12 (5.9%)	17 (13.9%)	6 (22.2%)	35 (9.9%)		
Meat	46 (22.4%)	16 (13.1%)	3 (11.1%)	65 (18.4%)		
Fruits and Vegetables	37 (18.0%)	16 (13.1%)	12 (44.4%)	65 (18.4%)		
Medical Equipment	92 (44.9%)	66 (54.1%)	2 (7.4%)	160 (45.2%)		

**Table 8 microorganisms-11-01951-t008:** Risk factors for MDR *E. coli* isolates from both clinical and environmental sources.

Variables	*p*-Value	AOR	95% CI for AOR	
			Lower	Upper
Clinical				
Pus	0.001	4.6	1.9	11.3
Male sex	0.010	2.1	1.2	3.9
Environment				
Water	0.019	2.6	1.2	5.8

AOR: adjusted odds ratio; CI: confidence intervals.

## Data Availability

All data generated during the study are presented in this paper and its Appendix A.

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
