# Peer review of "Antimicrobial Resistance Patterns and Risk Factors Associated with ESBL-Producing and MDR Escherichia coli in Hospital and Environmental Settings in Lusaka, Zambia: Implications for One Health, Antimicrobial Stewardship and Surveillance Systems"

_microorganisms, 2023, doi:10.3390/microorganisms11081951_

Round 1

Reviewer 1 Report

The study titled "Antimicrobial Resistance Patterns and Risk Factors Associated with ESBL-Producing and MDR Escherichia coli in Hospital and Environmental Settings in Lusaka, Zambia: Implications for One Health, Antimicrobial Stewardship and Surveillance Systems" is significant because it sheds light on the growing problem of antimicrobial resistance, specifically in ESBL-producing and MDR Escherichia coli, in both hospital and environmental settings in Lusaka, Zambia. The study identifies risk factors associated with the presence of these antibiotic-resistant bacteria and highlights the need for a One Health approach, as well as improved antimicrobial stewardship and surveillance systems, to combat this global threat.

Overall, the manuscript is well organized, well written and the topic is generally intriguing. The article contains few typographical and grammatical errors which need to be fixed before possible consideration.

I have few concerns:

> Introduction section is too lengthy and need to be concise.

> Upon reviewing the Results and Discussion sections, it appears that the results were not presented based on statistical analysis. Merely discussing the results based on percentage values, without taking into account the statistical analysis, lacks scientific rigor and will not provide a meaningful contribution to the scientific community.

> Discussion: In addition to the discussion based on statistical analysis, it is also essential to focus mainly on the significant findings of the study."

English quality is fine

Author Response

Dear reviewer 1,

Thank you very much for helping us improve our work.

Much appreciated.

Reviewer 2 Report

The article submitted for review argued that surveillance and routine screening for MDR and ESBL-producing is important to control the spread of resistant strains as part of a one-health approach. This research is critical  in the surveillance of AMR across the human, animal, and environmental sectors.

I found this article very interesting.
In my opinion the manuscript is well organized. The problem statements agree with the title and have significance for the research community. The methods used to gather the data for this article were clearly explained.  The citation quality is good. The presented literature review is impressive. The topic is interested and the result are concreted and useful for the scientific community. The presented research raises concerns, but at the same time suggests the need for supervision and control. I am counting on such a development of the situation and describing it in the publication.
However, I have a few comments:
1. Please explain all abbreviations in the article when using them for the first time.
2. In my opinion, the classification of pathogens determined during the research should be sorted out in the introduction. The description, of course, contains detailed data, but the introduction lacks a general classification of pathogens.
3. What does the term "as a national tertiary referral hospital" mean? Please explain.
4. Please explain why: all lactose fermenting colonies were considered to be E. coli

Thank you for considering my opinion. I encourage the authors to continue working on improving the manuscript.

Author Response

Dear reviewer 2,

Thank you very much for the good points and for helping us improve our paper.

Much appreciated and greetings.

Round 2

Reviewer 1 Report

The updated version of the manuscript remains deficient in terms of incorporating statistical analysis in both the presentation of results (textual and tabular data) and the discussion sections. The absence of statistical methods and analysis in these sections limits the robustness and rigor of the study, as statistical techniques are crucial for accurately interpreting and evaluating the findings. By neglecting the application of statistics, the manuscript fails to provide a comprehensive analysis and undermines the scientific validity and reliability of the research outcome.

The attached article can be of assistance to you in this regard.

English quality is fine

Author Response

Dear reviewer,

Thank you very much for your comments.

We have attached our responses.

Greetings from our team.
